# Human Postprandial Nutrient Metabolism and Low-Grade Inflammation: A Narrative Review

**DOI:** 10.3390/nu11123000

**Published:** 2019-12-07

**Authors:** Emma C.E. Meessen, Moritz V. Warmbrunn, Max Nieuwdorp, Maarten R. Soeters

**Affiliations:** 1Department of Endocrinology and Metabolism, Amsterdam University Medical Centers, Location Academic Medical Center (AMC), PO box 22660, 1100 DD Amsterdam, The Netherlands; e.c.meessen@amsterdamumc.nl; 2Department of Vascular Medicine, Amsterdam University Medical Centers, Location AMC, 1100 DD Amsterdam, The Netherlands; m.v.warmbrunn@amsterdamumc.nl (M.V.W.); m.nieuwdorp@amsterdamumc.nl (M.N.)

**Keywords:** postprandial inflammation, low-grade inflammation, nutrients, bile acids, microbiome

## Abstract

The importance of the postprandial state has been acknowledged, since hyperglycemia and hyperlipidemia are linked with several chronic systemic low-grade inflammation conditions. Humans spend more than 16 h per day in the postprandial state and the postprandial state is acknowledged as a complex interplay between nutrients, hormones and diet-derived metabolites. The purpose of this review is to provide insight into the physiology of the postprandial inflammatory response, the role of different nutrients, the pro-inflammatory effects of metabolic endotoxemia and the anti-inflammatory effects of bile acids. Moreover, we discuss nutritional strategies that may be linked to the described pathways to modulate the inflammatory component of the postprandial response.

## 1. Introduction

In modern times, humans spend more than 16 h per day in the fed state. The importance of the postprandial state has been acknowledged, since (postprandial) hyperglycemia and hyperlipidemia are linked with several chronic systemic low-grade inflammation diseases, including obesity, type 2 diabetes mellitus (T2D), atherosclerosis, non-alcohol fatty liver disease (NAFLD) and rheumatoid arthritis (RA) [1,2,3]. 

Inflammation can be defined as the primary reaction of the human body to deal with different kinds of infection or damage, which include swelling, redness, pain and fever [4]. Normally, this short-term adaptive response is a critical element of tissue repair and encompasses the integration of many complex signals in different cells, tissues and organs. However, the Western lifestyle induces a chronic state of systemic low grade-inflammation [5], which is involved in the pathogenesis of a wide range of non-communicable conditions such as obesity, metabolic syndrome, T2D, NAFLD and atherosclerosis [6,7,8].

The human immune system has two mechanisms to protect the host from pathogens, i.e., the innate (non-specific) and the adaptive immune system. The innate immune system is the first line of host defense and immune surveillance against pathogens. It acts independently of previous exposure to pathogens and consists of non-lymphoid tissue, mechanical barriers (e.g., mucosal epithelium) and cellular factors (e.g., neutrophils and macrophages). The activated innate immune system may also function as the initiator of chronic systemic low-grade inflammation, without any signs of local or systemic infection. The adaptive immune system (e.g., B- and T-cells) is dependent on the specific targeting of pathogens. As such, the adaptive immune system continuously builds on an immunological memory [9]. The cross-talk between nutrient metabolism and the immune system arises at many levels and varies from endocrine signaling to the direct sensing of nutrients by different immune cells [9]. 

In the postprandial state, the human body is exposed to high concentrations of a mixture of macronutrients, endocrine signals, gut-derived factors and others, with different plasma peak concentrations and time windows. Therefore, the postprandial state is dynamic and highly complex, and almost all organs and tissues are involved. Indeed, the different factors that make up a large part of the postprandial response subsequently influence metabolism, inflammation and health. 

Different nutritional patterns and macronutrients are associated with activation of the innate immune system, which in turn elicits systemic low-grade inflammation, which may contribute to the development of several diseases. Ingestion of a high-fat diet, high-carbohydrate diet, or the combination thereof causes postprandial inflammatory responses but also reactive oxygen species (ROS) generation [10,11,12,13,14]. Furthermore, consumption of a high-fat meal results in a temporary pro-inflammatory state via the so-called metabolic endotoxemia composing of bacterial wall products derived from gut microbiota [15,16]. In contrast, postprandial bile acid, released in the gastro-intestinal tract, is associated with anti-inflammatory effects [17,18,19]. 

The purpose of this review is to provide insight into the physiology of the postprandial inflammatory response, the role of different nutrients, the pro-inflammatory effects of metabolic endotoxemia and the anti-inflammatory effects of bile acids. Here, we focused mainly on human studies. Moreover, we discuss nutritional strategies that may be linked to the described inflammatory pathways to modulate the postprandial response. 

## 2. The Complex Postprandial Period as Framework in Low-Grade Inflammation

### 2.1. Three Metabolic States 

In general, the human body is subjected to three metabolic states: the postprandial state, the postabsorptive state and starvation [3,20,21]. The postprandial state, or the fed state, occurs after meal ingestion and embodies the digestion and absorption of nutrients (6–12 h), whereas the postabsorptive (i.e., fasted) state is the period when the nutrients are digested, absorbed, utilized and stored in the designated tissues (i.e., overnight). Starvation occurs rarely in healthy, well-nourished individuals, and is characterized by fat oxidation and ketogenesis to provide energy substrates for the brain [20]. Here, we focus on inflammation in the postprandial state, since the evidence for direct meal-induced inflammation is most compelling in this window.

Meal ingestion results in a complex and multifactorial (neuro)endocrine and metabolic response which influences postprandial inflammation via different pathways [22]. Western nutrition, high in calories, fats and refined sugars, results in an exaggerated increase of plasma glucose, triglyceride-rich lipoproteins (VLDLs), chylomicrons and their remnants [23].

### 2.2. Postprandial Period as an Endocrine, Metabolic and Inflammatory Frame-Work

Ingestion of a high-fat diet, a high-carbohydrate diet or a combination thereof, elicits postprandial inflammatory responses characterized with increased plasma lipopolysaccharides (LPS), interleukin-6 (IL-6), Tumor Necrosis Factor-α (TNF-α) levels and leukocyte counts in healthy subjects [11,12,13,14]. These meals also induced nuclear Factor kappa-light-chain-enhancer-of activated B cells (Nf-κB) binding activity [11,14] and reactive oxygen species (ROS) generation, whereas high fat ingestion alone also upregulated the expression of Toll-like receptors (TLRs) [11].

Several hormones are involved in the postprandial period and may influence postprandial inflammation. Most of the gut hormones, including bile acids, glucagon-like peptide-1 (GLP-1), fibroblast growth factor 19 (FGF19) and ghrelin, exert anti-inflammatory effects in the postprandial period [17,18,24,25]. In the white adipose produced, leptin predominantly elicits pro-inflammatory effects [26]. Insulin exerts both pro- and anti-inflammatory effects [27,28]. Furthermore, hormones involved in the hypothalamic pituitary adrenal (HPA) axis (i.e., adrenocorticotropic hormone and cortisol) are increased postprandial, and cortisol inhibits the production of several cytokines [29,30,31]. Therefore, the postprandial period can be defined as a complex endocrine, metabolic and inflammatory state (Figure 1). 

## 3. Nutrients

### 3.1. Hyperglycemia and Hyperinsulinemia 

Since the inflammatory response to meal ingestion is acknowledged as a physiological phenomenon, it is interesting to speculate which pathways are sequentially involved. Carbohydrates and lipid induce an inflammatory response in patients with obesity or T2D [32,33,34,35]. Indeed, saturated fatty acids can induce TLR4 mediated inflammation (see also 3.2, Fatty acids and TLR4 activation) [36]. High levels of these macronutrients, as well as insulin, affect the immune system [27,37,38,39,40]. Hyperglycemia elicits an increase in different cytokines, such as TNF-α, IL-1β and IL-6 [34,41,42], whilst insulin exerts the opposite effects. Insulin administration lowers CRP levels in patients with critical illnesses [40] and reduces intranuclear Nf-κB activation, ROS generation and monocyte chemoattractant protein 1 (MCP-1) levels in mononuclear cells [27]. In contrast, hyperinsulinemia is also associated with a pro-inflammatory phenotype. In healthy subjects and patients with obesity and/or insulin resistance, hyperinsulinemia activates the adipose tissue to produce TNF-α and IL-6 [28,38,43]. It is clear that hyperglycemia is associated with the pro-inflammatory state, which is also true for low-grade inflammatory conditions due to over-nutrition, where insulin exerts pro-inflammatory effects and presumably contributes to the pathogenesis of these diseases. On the other hand, hyperinsulinemia has anti-inflammatory effects during critical illness [40]. This might be explained by the profound severe inflammatory reaction under such circumstances.

### 3.2. Fatty Acids and TLR4 Activation

One of the defense mechanisms that evolved against microorganisms that can recognize pathogen-derived molecules such as LPS are called pathogen-associated molecular patterns (PAMPs) [44]. Receptors that can recognize PAMPs are called pattern recognition receptors (PRR), which can induce an innate immune response [44]. The three most important PRRs are toll-like receptors (TLRs), nucleotide-binding oligomerization domain (NOD-like receptors (NLRs) and retinoid acid-inducible gene-I (RIG-I) like receptors (RLRs)). Activation of these receptors can cause the release of pro-inflammatory cytokines such as TNF-α, IL-1β, IL-6 and IL-8 [44]. For activation of the innate immune system, TLRs are the most important receptors and LPS is the most important ligand of TLR4 [45,46], which explains the link between TLR4 activation in endotoxemic subjects [14]. However, saturated fatty acids have been shown to activate TLR2 and TLR4 and to provoke an inflammatory response in vitro [47]. Saturated fatty acids such as lauric acid and palmitic acid were shown to activate TLR4, whereas unsaturated fatty acids did not activate TLR4 [36]. This is explained by the antigenic part of LPS, which is called lipid A and usually consists of saturated fatty acids [47,48]. According to this, replacement of the saturated fatty acids in the lipid A domain with unsaturated fatty acids abolished the LPS-induced pro-inflammatory effect [47]. LPS- or saturated fatty acid-mediated activation of NF-κB, which is the downstream mediator of TLR4-activated inflammation [44], can be suppressed by the n-3 PUFAs docosahexaenoic acid and eicosapentaenoic acid [49]. The detailed interaction between fatty acids and TLR4 has already been reviewed elsewhere [46]. Other signalling pathways, such as bile acid signalling, also affect TLR4-mediated inflammation, as myeloid cell TLR4 activation can be attenuated by Farnesoid X Receptor (FXR) [50]. Conversely, activation of TLR4 in monocytes downregulates FXR [51], indicating a relationship between bile acid signalling and inflammation.

### 3.3. Amino-Acids

Remarkably, amino acids seem to have more anti-inflammatory effects in the postprandial period. For example, the amino acid glutamine inhibits Nf-κB [52] and also lowers LPS-stimulated TNF-α production in intestinal epithelial cells (rat) [53] and glycine inhibits oxidative stress in porcine intestinal cells [54]. Arginine inhibited the LPS-induced inflammatory response and oxidative stress in vitro [55]. In contrast, the dietary supplementation of arginine resulted in an altered intestinal microbiota which, in turn, resulted in the activation of intestinal immunity, including the upregulation of TLR4 and Nf-kB protein expression in mice [56]. Increased branched-chain amino acids (BCAAs) levels are associated with the development of insulin resistance [57], most likely in the presence of fat ingestion [58]. The BCAAs leucine, isoleucine and valine induced pro-inflammatory gene expression in the visceral adipose tissue of mice [59]. Additionally, peripheral blood mononuclear cells treated with BCAAs resulted in ROS production and activated Nf-κB supporting cytokine production, including IL-6 and TNF-α [60]. Postprandial amino acid metabolism and its role in postprandial inflammation is complex and not fully elucidated. Therefore, more mechanistic studies in humans are needed to determine the role of individual amino acids in human postprandial inflammation. 

## 4. Endotoxemia and Nutrients

### 4.1. Postprandial Endotoxemia 

Endotoxemia is defined as a state in which lipopolysaccharide (LPS), derived from the intestinal gut microbiota, is found in the circulation [61]. LPS is a protein that is a constituent of the cellular membrane of Gram-negative bacteria. Circulating LPS leads to increased systemic inflammatory markers in animal and human studies [16,62]. LPS levels are elevated after high-fat nutrition in mice and humans, and increase inflammatory markers in mice [15,16]. Increased postprandial circulating LPS levels are associated with increased inflammatory biomarkers such as IL-6 and soluble endotoxin receptor sCD14 in healthy subjects [63]. Healthy subjects following a high-fat and high-carbohydrate meal, compared to a meal rich in fruit and fibers, had increased plasma LPS levels and an increased expression of TLR2, TLR4, ROS and Nf-κB [14]. Furthermore, the increased LPS levels of obese mice and mice fed a high-fat diet could be lowered by oral antibiotic administration, suggesting a regulatory role for LPS levels by gut microbiota in mice [16]. In fact, changes in gut microbiota have been observed in animals and humans following a high-fat feeding [64]. Administration of oral vancomycin changed fasting LPS levels but did not affect postprandial plasma LPS levels and leukocyte levels in lean and obese subjects [12]. Direct administration of LPS in humans induced systemic inflammation and also insulin resistance, possibly due to an increase in cortisol and growth hormone [65,66]. According to this, a chronic overfeeding intervention of eight weeks in healthy subjects increased postprandial endotoxemia to a high-fat meal [67]. It has been proposed that the high-intake of omega-6 (n-6) polyunsaturated fatty acids (PUFA) in Western diets contributes to increased metabolic inflammation, as n-6 PUFA results in higher productions of eicosanoids such as leukotrienes, prostaglandins and lipoxins than n-3 PUFA [68]. A relatively high intake of n-6 PUFA compared to n-3 PUFA has also been suggested to increase inflammation, as n-6 PUFA would be used as a precursor of pro-inflammatory leukotrienes [69].

### 4.2. Lipopolysaccharide Translocation

It has been suggested that LPS can enter the body by two mechanisms. First, increased intestinal permeability facilitates the translocation of LPS to the portal vein. Secondly, LPS can be incorporated into new chylomicrons before it enters the lymphatic system and, eventually, the circulation [70,71]. High-fat-diet-induced microbiota change, with a reduction in *Lactobacillus* spp., *Bacteroides-Prevotella* spp. and *Bifidobacterium* spp. with parallel increased intestinal permeability, increased circulating LPS and inflammatory markers in mice [16]. An increase in intestinal permeability occurred through LPS application in vitro, which increased tight junction permeability via a TLR4-dependent process [72]. In a similar study design, a high-fat diet led to the depletion of intestinal eosinophils and increased permeability. Therefore, it has been hypothesized that the depletion of eosinophils induced by high-fat meals in mice is a result of an immune deficiency due to nutritional deficiencies induced by the high-fat nutrition [73]. These nutritional deficiencies are important; since eosinophils are abundant in the intestinal lamina propria in healthy subjects and play a role in gut mucus layer maintenance and immune homeostasis, depletion could contribute to increased intestinal permeability which is also seen in endotoxemia [74]. In humans, increased endotoxemia is already seen in healthy subjects following a high-fat meal [75]. In addition to this, infusion of Intralipids before LPS administration enhances the inflammatory response in healthy subjects [38]. However, conflicting data exist about this, as the study of Genser et al., only found increased intestinal permeability after application of lipids on collected jejunal tissue and not in vivo in humans with obesity and T2D [76]. Variable study results could be explained by different host responses to a high-fat diet, which could be determined by intestinal homeostasis factors such as immune state, mucus layer and gut microbiota composition. The host response to dysbiosis can differ in critically ill patients depending on the gut microbiota composition [77]. 

## 5. Other Players in Inflammation 

### 5.1. Transcription Factor Nf-κB in Postprandial Inflammatory Signaling 

As mentioned above, one of the (crucial) molecular drivers of postprandial inflammatory signaling in cells is Nf-κB. Nf-κB is a pleiotropic transcription factor and belongs to the primary “rapid acting” transcription factors. Therefore, Nf-κB is the first responder to harmful stimuli in the human body and, when activated, translocates from the cytoplasm to the nucleus [78,79]. However, meal ingestion itself induces Nf-κB activity, for example, in human mononuclear cells, and is therefore linked to postprandial inflammation [80]. Nf-κB activation leads to the gene expression of different cytokines (i.e., IL-6 and TNF-α), leukocyte adherence and chemotaxis [81,82]. On the other hand, cytokines, but also ROS and LPS, are acknowledged inducers of Nf-κB [82,83]. Macronutrients alone induce Nf-κB activity. Glucose ingestion increases intranuclear Nf-κB binding and TNF-α mRNA expression [84], and ingested carbohydrates with higher glycemic indexes induce higher Nf-κB activation in healthy lean subjects [85]. Additionally, fat ingestion also increases Nf-κB postprandial, but was not accompanied by an increase in inflammatory markers [86,87]. Little is known about the effects of amino acid ingestion on Nf-κB.

### 5.2. Oxidative Stress and Reactive Oxygen Species Production 

ROS are mainly produced in the mitochondria, plasma membranes, endoplasmatic reticulum and the peroxisomes via different mechanisms [88]. Nutrient availability results in an increase in oxidative stress, which is accompanied by higher ROS production [10,11,14,89,90]. Oxidative stress is described as an imbalance between oxidants and antioxidants. In favour of the oxidants (for example, ROS), postprandial oxidative stress results in disrupted redox signalling. Ingestion of a high-fat and/or carbohydrate meal, which results in temporary hyperglycemia and hyperlipidemia, prompts oxidative stress, which seems to be more extended in subjects who are obese or insulin-resistant [34,89,91]. Hence, different macronutrients affect the redox balance and postprandial oxidative stress. For example, in the peroxisomes, enzymes involved in postprandial free fatty acid β-oxidation and amino acid oxidation generate ROS as a result of their activity [92]. Additionally, glucose, lipid and protein ingestion induce ROS generation via mononuclear and polymorphonuclear leukocytes [89,90]. Furthermore, glucose ingestion increases the intranuclear binding activity of Nf-κB in monocytes, accompanied by an increase in ROS [10].

### 5.3. Complement Component Factor 3 Is Activated in the Postprandial State

The complement system is a part of the innate immune system and augments antibodies and phagocytic cells in defense against pathogens, and complement component factor 3 (C3) is an important activator of the complement system [93]. The complement system is activated in the postprandial state and chylomicrons are potent activators of C3 [94,95]. C3 is an important precursor of the Acylation Stimulating Protein (ASP). ASP is increased in the postprandial period and involved in fat metabolism, where it aids triglyceride clearance and fatty acid uptake in adipocytes [96]. After the consumption of fat, digested lipids are converted into triglycerides and packed into chylomicrons in the intestine. Chylomicrons deliver lipids to the peripheral tissues (i.e., muscle and adipose tissue). Chylomicrons are competent activators of C3 production in adipocytes [94]. Postprandial C3 levels increase after an oral fat load in humans and are associated with postprandial lipemia, whereas increased fasted C3 levels are associated with insulin resistance, hypertension, obesity, and coronary artery disease and its risk factors [95]. Van Oostrom et al. demonstrated that, when glucose is added to the oral fat load, the postprandial increase in C3 levels is prevented, and hypothesized that there is no peripheral free fatty acid trapping due to the lack of insulin after the oral fat load [97]. Furthermore, C3 is an independent risk factor for the development of metabolic syndrome [98] and dietary fat intake and its composition (high total dietary fat, saturated fatty acids (SFA) and mono-unsaturated fatty acids (MUFA)) are associated with an increased risk of the development of metabolic syndrome [99]. However, mice that lack C3 have delayed postprandial triglyceride clearance [96]. These data suggest that, in the acute setting of the postprandial state, C3 is necessary for the clearance of fat, but, in the case of of over-nutrition, may result in low-grade inflammation.

## 6. The Anti-Inflammatory Effects of Postprandial Bile Acid Signaling

### 6.1. Bile Acid Physiology

Besides their known function as emulsifiers, bile acids act as hormones via different bile acid receptors in postprandial energy metabolism. In humans, the primary bile acids (cholic acid and chenodeoxycholic acid) are synthesized from cholesterol in the hepatocyte. These bile acids can be conjugated to glycine or taurine, which enhances their solubility. Bile acids are then stored in the gallbladder [18]. After nutrient ingestion, cholecystokinine (CCK)-induced gallbladder contraction results in the release of bile acids in the duodenum, where they facilitate fat and fat-soluble vitamin digestion. The fat content and composition (long chain fatty acids > median chain fatty acids) correlates with bile acid secretion [100], and, therefore, the fatty acid composition of the meal influences bile acid release. When cholic and chenodeoxycholic acid reach the colon, they can be converted by the gut microbiota (via dehydroxylation) into the secondary bile acids, lithocholic acid and deoxycholic acid, or deconjugated. Most of the bile acids are actively reabsorbed in the ileum and a smaller part is passively reabsorbed in the colon [18]. Bile acids enter the liver via the portal vein, and the efficient enterohepatic cycle is complete [101]. Only 5% of the bile acids are excreted in the feces on a daily basis. Bile acids are mostly present in the enterohepatic cycle and reach concentrations that can activate their receptors. However, a fraction of those bile acids escape this hepatic uptake and reach the systemic circulation, where they activate different bile acid receptors in the systemic circulation [102]. Several nuclear and membrane bile acid receptors are involved in postprandial inflammation, including the Farnesoid X Receptor (FXR), Takeda G-protein-coupled receptor 5 (TGR5), Vitamin D receptor (VDR) and Pregnane X receptor (PXR) [17,18,19,103].

### 6.2. Farnesoid X Receptor and Fibroblast Growth Factor 19 

FXR was the first discovered bile acid receptor [104,105] and is expressed in liver, adrenals and the kidney [17,104,105]. Moreover, FXR is found in the small intestine, but with the highest expression at the ileum, which is the location where most bile acid reabsorption occurs [106,107]. The hepatic bile acid-activated FXR inhibits bile acid synthesis via the inhibition of CYP7A1 [108]. Additionally, FXR stimulates bile acid secretion, whereas it prevents bile acid reabsorption in the intestine and liver. Intestinal FXR activation results in the synthesis of the enterokine fibroblast growth factor 19 (FGF19). Some effects of the activated FXR on postprandial metabolism are facilitated by FGF19 [17]. FGF19 reaches the liver via the portal vein, binds to the hepatic FGFR4, and exerts effects on bile acid (decreases bile acid synthesis via CYP7A1) and energy metabolism (lowers gluconeogenesis and lipogenesis) [109,110].

FXR activation exerts several anti-inflammatory effects via different pathways (Figure 2). First in immune cells, FXR activation suppresses interferon gamma (IFNγ)-related genes in the macrophages [111]. Furthermore, in different mouse and human immune cells, FXR activation inhibited TNF-α production [112]. Second, treatment with FXR ligands results in the upregulation of the FXR reporter gene, inhibits Nf-κB activity and inhibits the pro-inflammatory enzyme inducible nitric oxide synthase (iNOS) in vascular smooth muscle cells [113]. Third, FXR activation maintains intestinal barrier integrity (less goblet loss, preserved intestinal barrier), and induces antibacterial gene expression in mice with induced colitis [112]. Fourth, FXR activation decreases endoplasmatic reticulum (ER) stress-induced NLRP3 inflammasome activation, assessed with IL-1β levels [114]. Finally, FXR ligands increase the gene expression of C3 in vitro and in vivo (rodents) [115], which links bile acid signaling to the complement system.

FXR not only has anti-inflammatory effects, but is also affected by the inflammatory response itself (Figure 2). Inflammatory stimuli (TNF-α and IL-1β) inhibited FXR transcription activity, most likely due to the upregulation of Nf-κB expression [116]. Moreover, as mentioned briefly in Part 4, TLRs modulate FXR gene expression. The activated membrane TLR4 inhibits FXR expression in human monocytes [51]. Macrophages treated with IFNγ inhibit FXR gene expression [111]. The postprandial state is accompanied by a tightly regulated balance between FXR mediated suppression of the inflammatory response and inflammation mediated inhibition of FXR. This balance cannot prevent chronic low-grade inflammation caused by chronic high-fat, high-glucose Western food habits. 

In vitro, mouse colonic epithelial cells which are pre-treated with FGF19 are protected against ROS (H_2_O_2_) [117]. Conversely, oxidative stress induces FGF19 mRNA expression in vitro, but not in vivo [118]. Furthermore, also in vitro, the activated FGFR4 by FGF19 inhibits Nf-κB signaling [119]. 

Macronutrients themselves influence FGF19 excursions. Carbohydrate ingestion elicits the fastest and highest increase of plasma FGF19 levels compared to fat or protein ingestion in healthy subjects [120], whereas the amount of increase in plasma bile acid levels is associated with fat ingestion [120,121]. The fact that carbohydrate ingestion elicits the highest concentrations of FGF19 may be explained by the fact that FGF19 inhibits hepatic gluconeogenesis and stimulates glycogen synthesis [109,110]. As such, FGF19 aids in the metabolic switch between glycogen synthesis and breakdown. 

### 6.3. Takeda G-Protein-Coupled Receptor 5 

In the postprandial state, TGR5 activation by bile acids stimulates the release of postprandial intestinal GLP-1 which, in turn, leads to the pancreatic secretion of insulin [122]. Furthermore, GLP-1 has several anti-inflammatory effects, including the inhibition of Nf-κB [25,123] TGR5 is expressed in many human tissues, including the hepatic Kupffer Cells (KCs), macrophages, intestinal L-cells, cholangiocytes, and the spleen, but not in hepatocytes [103]. TGR5 activation in hepatic macrophages and macrophages derived from peripheral blood inhibits phagocytic activity and production of pro-inflammatory cytokines (TNF-α, IL-1β, IL-6), whereas it hampers CD14/TLR4 activity and the Nf-κB pathway [124,125]. Bile acid-activated TGR5 induces the differentiation of IL-12 hypo-producing dendritic cells from monocytes, which promotes the immune response mediated by type 1 T helper cells [126]. In macrophages, the TGR5 agonist BAR501 results in the shift from macrophage polarization from the pro-inflammatory M1 to the anti-inflammatory M2 phenotype. This shift improved colitis in mice via the TGR5-dependent cAMP binding to the IL-10 promotor [127]. Moreover, bile acids inhibit NLRP3 inflammasome activation via the TGR5-cyclic AMP-protein kinase axis in mice [128]. Surprisingly, the concomitant inhibition of FXR and TGR5 stimulates the development of atherosclerosis [129]. The dual-specific FXR and TGR5 agonist INT-767 reduces monocyte infiltration and reverses obesity, hypercholesterolemia, NAFLD, and atherosclerosis in mice [130]. Therefore, postprandial TGR5 activation exerts several anti-inflammatory effects, predominantly via inhibition of the activated innate immune system via decreased cytokine production, Nf-κB activity, and the TLR4 pathway. 

Two other receptors—Vitamin D receptor (VDR) and the Pregnane X receptor (PXR), which are non-specific bile acid sensors—may also play a role in restraining the inflammatory response. VDR can be activated by the unconjugated form of the secondary bile acid LCA, which inhibits Th1 activation (adaptive immunity) in vitro [131]. Additionally, LCA downregulates Nf-κB activity and lowers IL-8 via IL-1β in a VDR-dependent fashion in colonic cancer cells [132]. Moreover, PXR inhibits gene expression of Nf-κB target genes, including TNF-α, and the generation of cytokines, whereas Nf-κB activation inhibits PXR [133].

## 7. Acute Postprandial Inflammation: A Physiological Phenomenon? 

The acute postprandial inflammatory response can be considered a physiological phenomenon. For example, postprandial-produced IL-1β in intestinal macrophages stimulates insulin release. Consequently, IL-1β and insulin promote the peripheral glucose disposal and stimulate the uptake of glucose by immune cells [134]. In addition to this, an inflammatory response initiated by a high-fat meal could also be physiologic. After absorption of fats from the intestine, chylomicrons and lipoproteins enter the circulation to be taken up by peripheral cells. If a relatively high number of chylomicrons or chylomicron remnants remain in the circulation, leukocytes can be activated to remove chylomicrons or their remnants from the circulation, inducing a physiological inflammatory cascade. However, if this process is out of balance, this leukocyte activation can also result in endothelial dysfunction and atherosclerosis [22]. Another example of a physiological aspect of postprandial inflammation is that ROS production, in low or moderate concentrations, is beneficial for humans [135]. Hence, phagocytes synthesize and store ROS, and use ROS against invading pathogens [136]. Nitric oxide functions as a signaling molecule between cells to modulate blood flow, and plays a pivotal role in the innate immune system via the elimination of intracellular pathogens [137]. Thus, the acute postprandial inflammatory response after meal ingestion seems to be a protective response to antagonize the potential harmful effects of macronutrients under healthy circumstances. An additional mechanism may be that the ensuing insulin resistance facilitates biosynthetic pathways, such as the pentose phosphate pathway, that support lipogenesis and, therefore, the storage of surplus nutrients [20].

However, being in the postprandial state for more than 16 h per day may result in a continuously activated immune system, ending in chronic low-grade inflammation. The activated immune system may support the development of low-grade inflammatory diseases.

## 8. Role of Nutrition in Chronic Low-Grade Inflammation Conditions Such as Rheumatoid Arthritis

### 8.1. General Interventions and the Gut

Because of this special issue of “Nutrients”, we incorporated some evidence on rheumatoid arthritis (RA). As the gut microbiome and food components, such as saturated fatty acids, can activate inflammatory pathways through TLR4 and Nf-κB [47], the role of nutrition and nutritional patterns in chronic low-grade inflammation has been the focus of several studies. In a Finnish study with diabetes mellitus type 1 participants [138], three nutritional patterns could be identified which were associated with lower LPS levels, such as high fish intake, frequently healthy snack consumption (fruits and fresh vegetables), and nutritional habits with a mix of poultry, fresh vegetables, pasta, meat dishes, grilled and fried food. However, they did not find an association between macronutrient, energy density, fibre intake and LPS levels [138]. In a study with obese or overweight subjects and low fruit, vegetable and whole grain intake [139], nutrition that was rich in whole grains or fruits and vegetables significantly decreased LPS biomarker LPS binding protein (LBP) levels at baseline. The vegetable and fruit group also decreased IL-6 levels at baseline, whereas whole grain consumption reduced TNF-α levels. Furthermore, the baseline proportions of *Clostridiales* of the phylum Firmicutes correlated with LBP changes after the nutritional intervention, with subjects with lower Bacteroidetes and higher Firmicutes showing a greater LBP decrease during the interventions [139]. Interestingly, in mice studies, a high abundance of *Clostridiales* has been associated with the development of RA [140], suggesting that microbiota and endotoxemia are associated with each other.

Another nutritional intervention which is associated with an improved clinical outcome is the Mediterranean diet [141]. Among Mediterranean diets, the Cretan diet has been proposed to be especially beneficial, perhaps due to the high intake of fresh vegetables, fruit, legumes and cereals [142]. Questionnaires evaluating the inflammatory activity and physical function of RA patients on stable drug treatment showed improved outcomes after a 12 week nutritional intervention, compared to control groups [143]. Also, the Mediterranean diet beneficially alters gut microbiota composition and plasma metabolites [144]. A meta-analysis focusing on the effect of the Mediterranean diet intervention on inflammation and endothelial function found that adherence reduced inflammatory markers CRP, IL-6 and intercellular adhesion molecule 1 (ICAM-1) [145]. As the Mediterranean diet has been shown to have beneficial effects for RA and is rich in n-3 fatty acids, phytochemicals, unrefined carbohydrate and oleic acid, other studies have also focused on the effects of single nutrients on RA, which is reviewed in detail elsewhere [141]. 

### 8.2. Omega-3 Fatty Acids

As described above, certain saturated fatty acids initiate an inflammatory response via TLR4 activation, whereas inflammation can be suppressed by the n-3 PUFAs. Indeed, there are two PUFA families, n-6 and n-3, which are essential fatty acids [46]. Both the ingested quantity and the n-6 and n-3 ratio are important for metabolic health, as humans evolved on a n-6/n-3 ratio of around 1. Western diets have a ratio of around 15/1 [146]. A high n-6/n-3 ratio is associated with several diseases such as cardiovascular diseases, cancer, inflammatory and auto-immune diseases [146]. On the contrary, nutritional intake with a high n-3 fatty acid intake has been suggested to result in lower incidence of cardiometabolic diseases [69]. The proposed underlying mechanism to this is the competition between n-6 and n-3 as a precursor for prostaglandin formation, and the use of n-3 fatty acids results in lower pro-inflammatory leukotrienes [68]. In fact, a meta-analysis evaluating the use of fish oil in RA found a reduction in joint tenderness and morning stiffness in the initial analysis; after correction for confounders, only joint tenderness remained significantly improved [147]. A more recent meta-analyses evaluated the pain-reducing effect of n-3 fatty acids on patients with rheumatoid inflammatory joint pain or joint pain secondary to inflammation. In these meta-analyses, a reduction in joint pain minutes, intensity, number of painful joints, non-steroid anti-inflammatory drugs (NSAID) use and morning stiffness was found, whereas only one meta-analysis found a reduction in leukotriene B4 [148,149,150]. The strongest effect of n-3 fatty acids was seen with a non-olive oil intake of > 3 g/day [148,149]. Accordingly, in the study of Lyte et al., healthy individuals following a saturated fatty acid meal had increased endotoxemia and inflammatory parameters, which decreased after a meal rich in n-3 PUFA, but not after a meal rich in n-6 PUFA [151]. However, a Malaysian crossover study where participants followed a five week nutritional intervention with saturated fatty acids, monounsaturated fatty acids and a PUFA intervention, did not find differences in inflammatory markers, perhaps due to the relative high energy intake in all interventional groups, or differences in the response to specific nutritional components between populations [152]. 

### 8.3. Fasting Paradigms

Another strategy to influence LPS levels is caloric restriction and fasting. A four week regimen of 800 kcal per day in obese women decreased LBP, CRP, gut permeability as well as insulin resistance [153]. In the observational study of Toledo et al. [154], healthy subjects and subjects with pre-existing conditions followed a fasting regimen with a low caloric intake of around 225 kcal for 4 to 21 days. From the group with pre-existing conditions which included inflammatory diseases, 84% reported clinically relevant improvement in their symptoms after fasting [154]. According to this, repeated cycles of three day fasting improved symptoms and alleviated demyelinisation in mice from an experimental model mimicking multiple sclerosis, which is, like RA, an immune-mediated inflammatory disease [155]. Furthermore, 20% of these mice did not have any symptoms after the diet. This effect was associated with increased regulatory T cells and corticosterone levels, which decreased the pro-inflammatory response [155]. In RA patients, CD4+ T-cell activation with subsequent differentiation into Th1 and Th17 has been suggested to be responsible for RA progression [156]. Furthermore, expression of TLR4 on CD8+ T cells was correlated woth RA severity [157]. Interestingly, a seven day fasting regimen resulted in a decrease in CD4+ and CD8+ lymphocyte levels and an increase in IL-4, which is known to have anti-inflammatory effects [158,159]. Clinical improvements were found after a seven day total fasting regimen, such as decreased morning stiffness, modified Lansbury articular index and erythrocyte sedimentation rate [160]. 

In the study of Kjeldsen-Kragh et al. [161], patients followed a 7–10 day fasting regiment with 800 kcal/day without any fruit juices followed by a vegan diet for 3–5 months and a subsequent vegetarian diet intervention for 7–9 months. Improvements in inflammatory markers and RA symptoms after the fasting were maintained until the end of the study compared to the control group [161]. In fact, a fasting regiment of 7–10 days resulted in decreased CD4+ cells and CD4+ cells were inversely correlated with cortisol levels in peripheral blood, however, this study did not find differences in interleukins, although natural killer cell activity was higher [162]. The role of NK cells in RA is not clear, as both protective and detrimental roles have been described [163]. Interesting insights into the mechanism of decreased inflammation due to fasting comes from animal studies. The number of circulating leukocytes were decreased by 87% in hibernating ground squirrels [164]. Furthermore, mediation of T cell immunity was supressed after a three day fasting period in female Mongolian gerbils [165]. Fasting in humans decreased leukocyte counts by 0.4–1 × 10^3^ /μL depending on the length of fasting, from 5–20 days, respectively [154]. Whether these reductions in leukocyte count are sufficient to decrease the inflammatory effects of auto-immune diseases remains to be elucidated. However, a previous study only found temporary improvements of fasting with fruit–vegetable juices and no beneficial effect of a nine week vegan diet intervention in RA patients, suggesting that possibly more complex mechanisms, such as disease state or immunity state, should be considered in addition to the fasting regimen [166]. These studies indicate that several fasting regimens with subsequent nutritional adjustments can decrease RA associated symptoms, possibly by influencing the immune system. 

Although several fasting studies were effective for RA patients, we feel that it is premature to advise fasting. First, the optimal fasting regimen is not known. Also, healthy nutritional intake, as described above, reduces inflammatory activity and decreases symptoms. A potential complication of fasting protocols may be the ensuing loss of lean body (muscle) mass and malnutrition. Reduced caloric intake in combination with low protein ingestion and low exercise frequency can reduce lean body mass, which is associated with negative health implications [167]. Future studies should focus on identifying patient characteristics that predispose an effective and safe fasting intervention. More research in clearly stratified patient groups could provide more insights into the mechanism and scope of its effectivity in RA patients.

## 9. Conclusions, Implications and Future Perspective 

Humans spend more than 16 h in the postprandial state, acknowledged as a complex interplay between nutrients, hormones and metabolites. The postprandial state is influenced by nutritional components such as saturated and unsaturated fatty acids. A high-fat diet is associated with increased intestinal permeability and translocation of LPS into the circulation. LPS and dietary components such as saturated fatty acids have pro-inflammatory effects through TLR4 signaling. On the other hand, postprandial bile acids have anti-inflammatory effects via the enterohepatic activation of FXR, TGR5 and VDR. Dietary interventions such as the Mediterranean diet or fasting have been shown to ameliorate RA symptoms in some studies. However, we also show that clear, evidence-based, personalized advice is currently lacking, which might be seen as an unavoidable limitation of this paper. Therefore, a balanced healthy nutrition or hypocaloric nutritional intervention may be used as a treatment option that will synergize with pharmacological interventions. Larger prospective trials, that take inter-individual variability into account, are necessary to confirm current findings and move towards meaningful clinical and personal advice.

## Figures and Tables

**Figure 1 nutrients-11-03000-f001:**
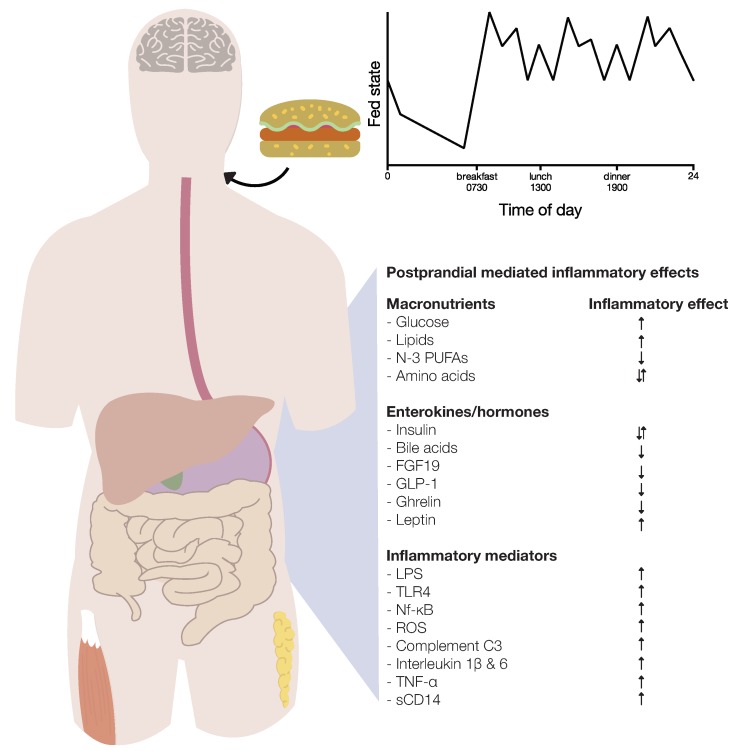
The postprandial period as an endocrine, metabolic and inflammatory framework. Meal ingestion results in a complex and multifactorial endocrine and metabolic response, which influences postprandial inflammation via different pathways. Ingestion of glucose and lipids induces postprandial inflammation, whereas amino acids have pro- and anti-inflammatory effects. During the postprandial period, the enterokines insulin, bile acids, fibroblast growth factor 19 (FGF19), glucagon-like peptide- 1 (GLP-1) and ghrelin are released and exert anti-inflammatory effects on postprandial metabolism. Leptin mediates negative effects in adipose tissue. Insulin also has pro-inflammatory effects. Furthermore, as a result of nutrient ingestion, several mechanisms (i.e., lipopolysaccharide (LPS), Toll-like receptor 4 (TLR4), nuclear factor kappa-light-chain-enhancer-of activated B cells (Nf-κB), reactive oxygen species (ROS), complement component factor 3 (C3), interleukins, Tumor Necrosis factor (TNF)-α and soluble CD14 are activated or produced, and stimulate postprandial inflammation.

**Figure 2 nutrients-11-03000-f002:**
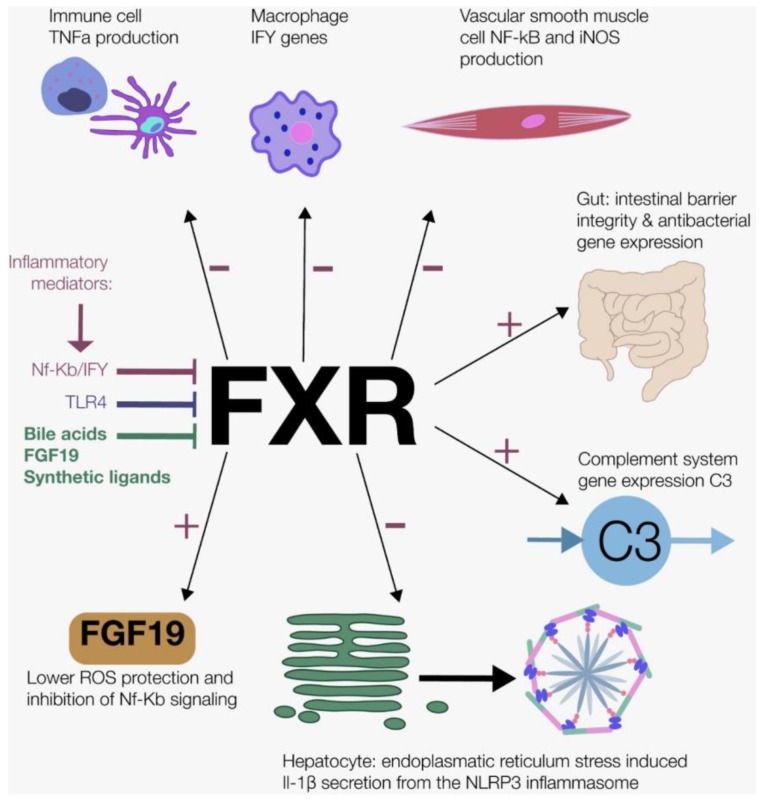
The role of Farnesoid X Receptor (FXR) in postprandial inflammation. FXR activation exerts several anti-inflammatory effects via different pathways: FXR activation suppresses interferon gamma (IFNγ)-related genes in macrophages and inhibits Tumor necrosis Factor-α (TNF-α) production in immune cells, inhibits nuclear factor kappa-light-chain-enhancer-of activated B cells (Nf-κB) activity and the pro-inflammatory enzyme inducible nitric oxide synthase (iNOS) in vascular smooth muscle cells, maintains intestinal barrier integrity and induces antibacterial gene expression, decreases endoplasmatic reticulum (ER) stress-induced NLRP3 inflammasome activation assessed with interleukin-1beta (IL-1β) and increases the gene expression of complement component factor 3 (C3). FXR not only has anti-inflammatory effects, but is also affected by the inflammatory response itself. Inflammatory stimuli TNF-α and IL-1β activate FXR via Nf-κB expression and IFNγ, Toll-like receptor 4 (TLR4), bile acids, fibroblast factor 19 (FGF19) and synthetic ligands also stimulate FXR.

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
