# Peer review of "Human Postprandial Nutrient Metabolism and Low-Grade Inflammation: A Narrative Review"

_nutrients, 2019, doi:10.3390/nu11123000_

Round 1
Reviewer 1 Report
well written and interesting review with innovative aspects
Author Response
We thank the reviewer for these comments.

Reviewer 2 Report
The Authors aimed to analyse the role of the postprandial inflammatory response, as well as the the role of different nutrients, the pro-inflammatory effects of metabolic endotoxemia and the anti-inflammatory effects of bile acids. They performed a narrative review. Nevertheless the should indicate the study`s design with a commonly used term in the title. The abstract provide an informative and balanced summary of what was done by the researchers. The topic is of the interest. The background is explained in a rationale way. THe methodology section is well described presenting the study design, the data sources and flow-chart. This research do not have a specific section of "Discussion". The authors should discuss the limtiations of the study, taking into account sources of potential bias or imprecision. Finally, the authors give a cautious overall interpretation of results considering objectives and they conclude in an honest way.
The research topic is of interest.
The study is well designed and written in an acceptable way. References are updated.
They conclude in an honest way.
Author Response
Dear Editor,
We thank the reviewers for the careful comments. We have changed the titel (added narrative review) and emphasized the limitations (although one of the reviewers complimented on the limitations in the paper) => line 534: "However, we also show that clear evidence based personalized advice is currently lacking, which might be seen as an unavoidable limitation of this paper."
Yours sincerely, Maarten Soeters

Reviewer 3 Report
Congratulations on a very thorough and well-written review of the interesting topic of association of health, nutrients and postprandial inflammation.
Very interesting review.
Clear style.
A comprehensive overview of the topic.
It addresses limitations and gaps in evidence.
Author Response

(The authors gave the same response as above.)
